# Accounting for the Nutritional Context to Correctly Interpret Results from Studies of Exercise and Sedentary Behavior

**DOI:** 10.3390/nu11092230

**Published:** 2019-09-16

**Authors:** Barry Braun, Alissa Newman

**Affiliations:** Department of Health and Exercise Science, Colorado State University, Fort Collins, CO 80523, USA

**Keywords:** energy balance, insulin sensitivity, glucose metabolism, exercise science, inactivity, experimental design

## Abstract

There is a wealth of research lauding the benefits of exercise to oppose cardiometabolic disease such as diabetes, CVD and hypertension. However, in the great majority of these studies, the nutritional context (energy balance, deficit, or surplus) has been ignored, despite its profound effect on responses to both exercise and inactivity. Even a minor energy deficit or surplus can strongly modulate the magnitude and duration of the metabolic responses to an intervention; therefore, failure to account for this important confounding variable obscures clear interpretation of the results from studies of exercise or inactivity. The aim of this review is to highlight key lessons from studies examining the interaction between exercise and sedentary behavior, energy status, and glucose and insulin regulation. In addition to identifying notable problems, we suggest a few potential solutions.

## 1. Introduction

The phrase “exercise is medicine” has gone from a useful analogy to a slogan that is broadcast far and wide; and has trickled down from the halls of health science departments to the mass media and general public. It is meant to denote that habitual exercise, like pharmaceutical compounds (or dietary strategies), can be used to potentially prevent, treat or mitigate disease and disability. A plethora of research studies have lauded the benefits of habitual exercise—defined as regular exercise performed on a regular basis over months, years or a lifetime—as a means to greatly reduce the risk for obesity and obesity-related metabolic disease (e.g., cardiovascular disease, Type-2 diabetes, hypertension, etc.), as well as a host of other health issues too numerous to include in this review. Increasingly, the value of short-term exercise (i.e., a single bout or a few bouts) has been recognized as providing benefits to metabolic health. Extending the “exercise is medicine” analogy, if each exercise bout is thought of as a dose, in much the same way as we think of a dose of a drug, then we can expect a beneficial change in metabolic function that lasts for a discrete period of time before the effects wane and another dose needs to be re-applied to maintain the health benefit. Basic pharmacology recognizes that the body’s nutritional milieu plays a key role in the response to a drug. Therefore, pharmaceutical drugs are approved with instructions as to whether they should be taken with or without food to get the appropriate/most potent effect and/or minimize side effects. Although there are no FDA-mandated instructions or stringent guidelines, the same is certainly true for exercise. It is not surprising to anyone reading this review that energy balance and dietary intake have a profound effect on the response to/tolerance of exercise.

The converse is also true; exercise can have a powerful effect on energy balance, both in the short- and long-term. Acute bouts of exercise do not typically elicit compensatory increases in hunger and/or food intake in the short-term [1]; in fact, very intense bouts of exercise can suppress hunger, a phenomenon known as “exercise-induced anorexia” [2,3]. This hunger suppression may lead to a negative energy balance that can last for several days [4,5,6,7]. A number of studies have shown that acute exercise tends to reduce appetite and lower the circulating concentrations of appetite-enhancing hormones like ghrelin [8,9,10]. Concurrently, there is often an increase in satiety and satiety-promoting hormones like GLP-1, GIP, and CCK [3,8,10]. Food intake, not only total energy consumed but also preference for macronutrients (protein, carbohydrate and fat) is modulated by prior exercise, although the direct relationship between intake and the appetite/satiety-regulating hormones is not always clear [11,12,13]. When exercise is regularly performed over a longer period of time, such as in a multi-week exercise training study, it may take weeks for energy intake and expenditure to reach a new equilibrium [1]. Importantly, compensatory increases in energy intake are not uniform and the compensation varies greatly between individuals [5,14,15]. In fact, individuals can lose significant weight, have little change or even gain body weight in response to the same training intervention, as shown in Figure 1 [16], illustrating the enormous inter-individual variability that confounds making blanket statements about the role for exercise in weight loss.

Therefore, interpreting results from exercise studies requires knowledge of the nutritional context, i.e., energy deficit, balance or surplus, in which they were performed. Since the most profound effects (and best evidence) come from studies of how exercise impacts insulin resistance and sensitivity in people with or at risk for prediabetes/diabetes, this review will be focused on the impact of energy imbalance on studies of exercise and glucose metabolism/insulin sensitivity.

## 2. Importance of Energy Balance

There are dozens of studies highlighting improvements in insulin and glucose regulation that result from even modest amounts of exercise [17,18]. In some of these studies, care is taken to control for caffeine intake, gender, age, training status, pharmaceutical or nutraceutical use and other factors (most recently, circadian rhythm) that are known or suspected to influence the hormonal response to exercise [19]. Energy intake, and the relationship between intake and expenditure (i.e., energy balance) rarely merits more than a few sentences, if that. When it is considered, that consideration is generally limited to collection of dietary recall records with little or no discussion of whether those records were collected in a manner that inspires confidence in their validity, what the results actually were or how those results influence interpretation of the main outcomes of the study. Often the extent of dietary control is to ask research participants to replicate their pre-intervention meal or meals before the post-intervention testing. That can be a viable strategy with stringent tracking or follow-up but there is rarely any discussion of whether the strategy was successful in replicating the pre-intervention nutritional context or even whether any attempt was made to determine whether participants actually did what they were asked to do. From the perspective of study logistics, controlling for dietary intake raises costs, increases workload for researchers and adds to the participant burden. A simple approach taken by many researchers is to assess participants’ body weight and assume that a lack of significant change over time is indicative of energy homeostasis. This approach is certainly easier than experimentally controlling for energy balance, but is problematic, particularly for shorter-term interventions. Blundell and colleagues [14] contend that the premise that body weight stability is proof of energy balance is a circular argument. The supposition is that body weight is stable because of energy homeostasis; however, the evidence for homeostasis of energy is the stability of body weight. Given the day-to-day variability in body weight and the potential for the ratio of lean and fat mass to change in response to an exercise intervention without clinically relevant changes in total weight, using stable body weight as proof of energy balance is fraught with peril.

The lack of strict attention to nutritional variables in most exercise studies is particularly surprising given that even minor energy deficits or surpluses can have a profound effect on the duration and magnitude of the metabolic responses to exercise [20]. Rather than provide an exhaustive survey of all the possible interactions between energy balance and exercise, in this review, we will focus on a few examples that serve to illustrate the larger issue. Because there are ample research data on the interactions between exercise and energy balance on insulin resistance and sensitivity, we will highlight some of the key lessons from those studies.

Notably, even in studies in which there is strict control of energy intake, the confounding influence of energy imbalance can still obscure clear interpretation of results. Investigators rarely, if ever, deliberately increase energy intake to offset the higher energy expenditure in studies composed of exercise and no-exercise conditions. In one of our own studies, women with mild type-2 diabetes completed three in-patient conditions at San Francisco General Hospital [21]. All meals were prepared and served by a metabolic kitchen based on energy requirements calculated by highly trained staff. In two of the conditions, participants expended 750 kcals over the course of two days by walking on a treadmill while in the other condition, they were sedentary. Energy intake was the same in both conditions. We found that exercise enhanced insulin sensitivity—measured using a glucose, insulin and stable isotope infusion—by about 25% relative to the no-exercise control. However, because energy intake was not increased to match the 750 kcal higher output in the exercise conditions, the potentially confounding effects of a modest energy deficit, which also enhances insulin sensitivity, cannot be ruled out. To directly address this issue, Black et al. did a study in which the additional energy expenditure from exercise was either ignored (as in the Braun et al. study) or matched by adding energy intake to maintain energy balance. They showed that 6 days of treadmill walking (500 kcal/day) improved peripheral and hepatic insulin action in overweight, sedentary subjects compared to their pre-intervention baseline. As shown in Figure 2, when the 500 kcal/day was fed back to the participants to maintain energy balance, the improvements were abolished [22]. Further, many other outcome measures (e.g., leptin, adiponectin, see Table 1) that changed in response to exercise without kcals being re-fed were unchanged from baseline when the “exercise energy” was added to the diet [21]. These findings have been corroborated by others who have found that insulin action is enhanced for 24–72 h post-exercise without replacement of energy expenditure [21,23,24], but when energy balance is restored, the effect on insulin may be attenuated or abolished [25,26].

## 3. Influence of Macronutrient Availability

While energy balance itself is clearly a confounding variable, the composition of post-exercise energy intake also appears to be important. Thirty years ago, Cartee and colleagues first demonstrated that post-exercise carbohydrate feeding reversed the exercise-induced enhancement of insulin action, whereas restricting carbohydrate maintained the effect for several days [23]. More recent studies have demonstrated similar results [27,28]. Animal studies involving high carbohydrate feedings following exercise have demonstrated a profound decrease in insulin responsiveness in both insulin- and contraction-mediated pathways [29,30] and it has been suggested that this is due to a decrease in GLUT4 translocation to the cell surface [31,32]. Prevention of glycogen supercompensation after glycogen-depleting exercise allows the increased GLUT4 expression and enhanced insulin action to persist for several days, which was reversed upon carbohydrate feeding [30].

Another way to test for the effects of energy versus macronutrient availability is to maintain energy balance but greatly increase the fat content of the diet [33]. While increased free fatty acid availability has been long been associated with reduced insulin sensitivity [34], there does not appear to be a significant effect on insulin sensitivity following exercise [35,36,37]. Shenck and Horowitz suggest that, following exercise, there is a preferential partitioning of fatty acids towards oxidation and storage as intramuscular triglyceride [38]. It is worth noting that other aspects of substrate metabolism may be altered in the presence of energy and/or fat deficit [33], but that is beyond the scope of this discussion. The relationship between protein surplus or deficit and insulin sensitivity after exercise has not been explicitly tested but it is reasonable to expect that, like fat, insulin sensitivity would be augmented as a consequence of displacing carbohydrate.

## 4. Energy Surplus

Like energy deficit, energy surplus has a rapid impact on insulin sensitivity but in the opposite direction, i.e., short-term energy surplus reduces insulin sensitivity [39,40]. Although there are fewer studies in which exercise energy has been *over*compensated by adding energy intake greater than the cost of the exercise, some trends are apparent. In one study, Hagobian and Braun [41] used 3 days of +750 kcal/day energy surplus and restricted exercise to impair insulin sensitivity (as estimated with a standard glucose tolerance test) in relatively lean healthy participants. On the next day, participants performed 750 kcal of exercise, but energy intake was further increased to maintain the 750 kcal energy surplus. The following day, the pre-intervention insulin action had been partially, but not completely restored (Figure 3), suggesting that the full exercise effect was blunted in the presence of energy surplus (but also suggesting that a single bout of exercise partially ameliorates the negative impact of energy surplus—a potentially practical piece of information for individuals who are experiencing positive energy balance during holidays or while traveling). Similarly, Walhin and colleagues found that 7 days of overfeeding and inactivity impaired insulin sensitivity and altered expression patterns of genes and proteins in adipose tissue that are associated with metabolism and insulin action. When combined with daily exercise, however, these effects were reduced or prevented entirely [42].

## 5. Inactivity, Energy Balance, and Insulin Sensitivity

In the past 10–15 years, there has been increasing attention on sedentary behavior and physical inactivity [43,44,45]. In this context, inactivity is not just a lack of the positive effects of being physically active, strong research has shown that prolonged sitting results in negative consequences to health (risk for Type 2 diabetes, obesity, cardiovascular disease, etc.) with distinct mechanistic underpinnings [44,46,47]. Akin to the situation with adding exercise, however, subtracting movement to create a “sedentary” research condition cannot be separated from the energy balance context. Studies of bed rest or prolonged sitting designed to mimic sedentary behavior are almost invariably confounded by positive energy balance unless strict care is taken to lower energy intake to match the low expenditure. The danger to clear interpretation of results is that “positive energy balance due to overfeeding is a confounding variable that exaggerates the deleterious effects of physical inactivity” [48].

In one of our own studies, we deliberately tested the hypothesis that the reduced insulin sensitivity observed after short-term sitting was partly attributable to energy surplus. Stephens et al. [49] tested whether a day of inactivity (15 h continuous sitting) would reduce insulin sensitivity measured the following morning in 14 healthy young individuals. Participants were tested in three conditions, an active control (no structured exercise but a lot of standing, walking and very little sitting–NO-SIT) with energy intake matched to expenditure; an inactive group with almost no activity (even restroom breaks were accomplished by having participants taken to the test room in a wheelchair) with the same energy intake as the control (resulting in a positive energy balance of 900 kcal–SIT) and a third group that also sat continuously but with energy intake slashed by 900 kcal to match the new low expenditure (meaning they were in energy balance–SIT-BAL). As seen in Figure 4, Stephens et al. reported that a day of sitting combined with energy surplus reduced insulin action by 39% relative to the active control condition, an effect that was blunted (but not prevented entirely, decrement was still −18%) by matching energy intake to expenditure [49]. Over a longer timeframe, Winn et al. [50] found that over the course of 10 days of relative inactivity (~4000 steps per day), individuals who were in positive energy balance exhibited increased body weight, body fat, and impaired insulin sensitivity. Individuals who were in energy balance or a slight deficit did not experience these metabolic impairments.

Bed rest is a unique model to investigate inactivity, as it combines inactivity with head-down tilt that is also a research model for microgravity. Similar to the above, bedrest is inextricably associated with energy surplus unless energy is deliberately matched to the very low expenditure. In one of the few studies to examine the effects of bed rest while tightly controlling for energy balance, Bergouignon et al. performed a 2-month bed rest study in women [51] and a 3-month bed rest study in men [52], in which diets were tightly controlled to maintain fat mass. The authors suggest that body fat stability is indicative of energy balance as body mass fails to account for bed-rest-induced muscle atrophy and subsequent fat mass increase, which is indicative of energy surplus [53]. Despite being in energy balance, the results of both studies suggest that prolonged bed rest does impair insulin sensitivity.

As noted earlier with respect to “exercise added” studies, macronutrient composition is a factor that matters with studies of inactivity. Stettler et al. [54] found a 24% decrease in insulin sensitivity after 60 h of bed rest combined with a high saturated fat diet (45% of energy from fat, of which 60% was saturated fat), but no change in insulin sensitivity after 60 h of bed rest and a high carbohydrate diet (70% of intake). This result is in stark contrast to the majority of studies focused on exercise and insulin sensitivity, in which restricting carbohydrate seemed to be necessary to prolong/exacerbate the insulin-sensitizing effects of prior activity. If the results by Stettler are confirmed, it adds further evidence to the hypothesis that the benefits of exercise and the harm of inactivity are regulated by fundamentally different mechanisms.

## 6. Conclusions

The objective of this brief review was to illustrate the critical importance of accounting for the energy balance “context” to correctly interpret results from studies of exercise or inactivity. In particular, ensuring participants (or non-human models like rodents, etc.) consume a diet in which energy intake is at least roughly, and ideally strictly, matched to expenditure is necessary. Energy imbalance likely confounds interpretation of results attributable to either exercise or inactivity because energy deficit exacerbates, and energy surplus attenuates, the “true” effect of the movement intervention. Further, the composition of energy intake in terms of macronutrients also modulates the effects of a movement intervention and care should be taken to match pre-intervention and post-intervention meals for at least 24–48 h prior to assessments. Given the reality that high costs and the administrative burden of creating and providing all meals for all participants may be prohibitive for many trials, creative alternatives can be employed. Rather than instructions to “eat the same diet”, participants can use technology (e.g., phone camera) to document their intake in the 24–48 h before testing and that information provided back to them by investigators before the post testing with intentional reminders and follow-up. A standard for adherence to the prescribed diet can be developed and participants who do not meet that standard should be dropped from the study (or possibly rescheduled for a second try). The consequences of continuing to minimize the necessity for strict control of energy balance are evident, as we are habitually confronted with confusing, opposing and hard-to interpret studies of exercise effects that are confounded with dietary effects.

Here, we have tried to use one concept (insulin sensitivity) that is strongly influenced by the control, or lack of control, of energy intake, as an exemplar of a bigger issue. Because there is a wealth of research in this area, and because it seems to be particularly influenced by dietary factors, we believe this was a reasonable choice. But there are numerous other areas of research to which this theme applies, several of which have been previously discussed by Braun and Brooks [55]. Additionally, although it adds more fuel to the “importance of understanding the context” fire, it is just as likely that sleep, circadian biology, stress and other physiological states modulate the effects of exercise or inactivity on health outcomes. We expect those areas are ripe for future reviews like this one that are intended to provide constructive criticism to move the field forward.

## Figures and Tables

**Figure 1 nutrients-11-02230-f001:**
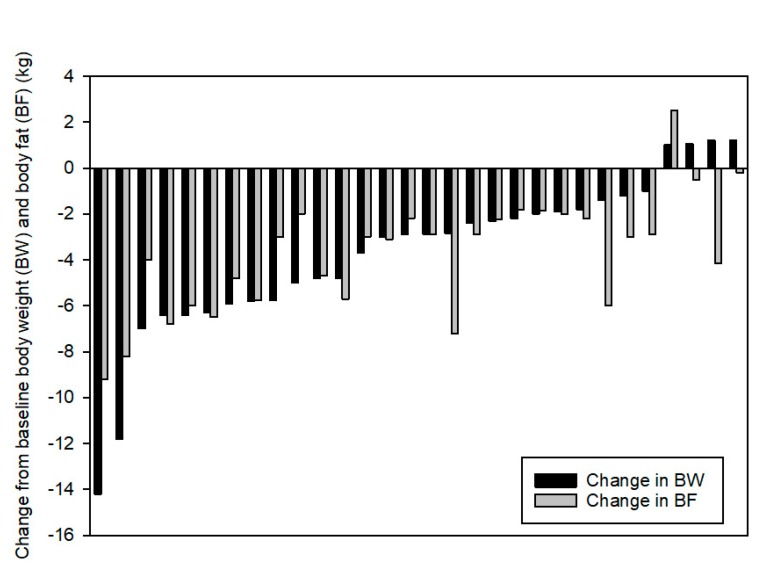
Individual body weight and fat mass changes following 12 weeks of supervised exercise. Each pair of histograms represents one participant. Reproduced with permission from Springer Nature: Springer Nature, International Journal of Obesity [16], Copyright (2008).

**Figure 2 nutrients-11-02230-f002:**
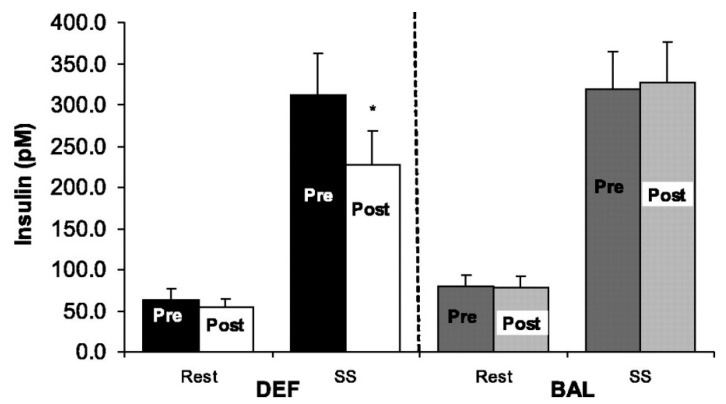
Change in resting and steady-state (SS) insulin concentrations in energy deficit (DEF) and energy balance (BAL). Values are means ± SE. * Significantly different from pretraining, *p* < 0.05. Reprinted with permission from the authors [22]; Copyright © 2005 the American Physiological Society.

**Figure 3 nutrients-11-02230-f003:**
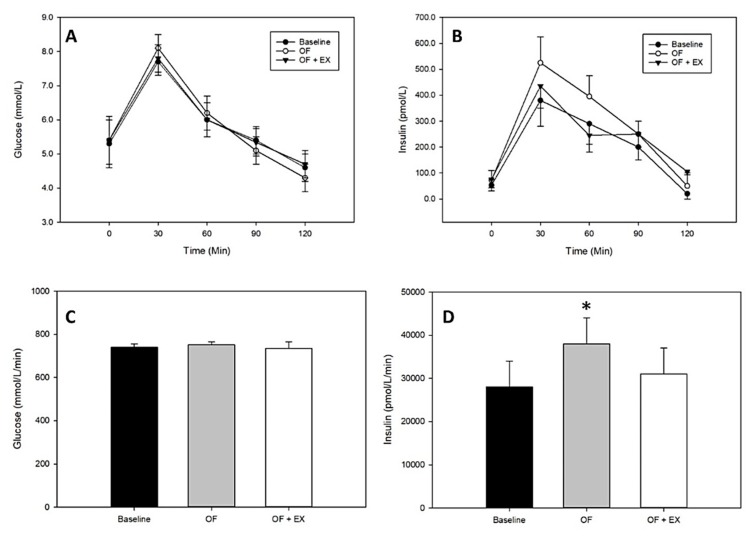
**Top** panels: glucose (**A**) and insulin (**B**) responses during a 2 h oral glucose tolerance test. **Bottom** panels: glucose (**C**) and insulin (**D**) areas under the curve. OF: overfeeding condition; OF + EX: overfeeding plus exercise condition. Values are mean ± SEM. * Significantly different from baseline, *p* < 0.05. Adapted from Hagobian and Braun, 2006 [41].

**Figure 4 nutrients-11-02230-f004:**
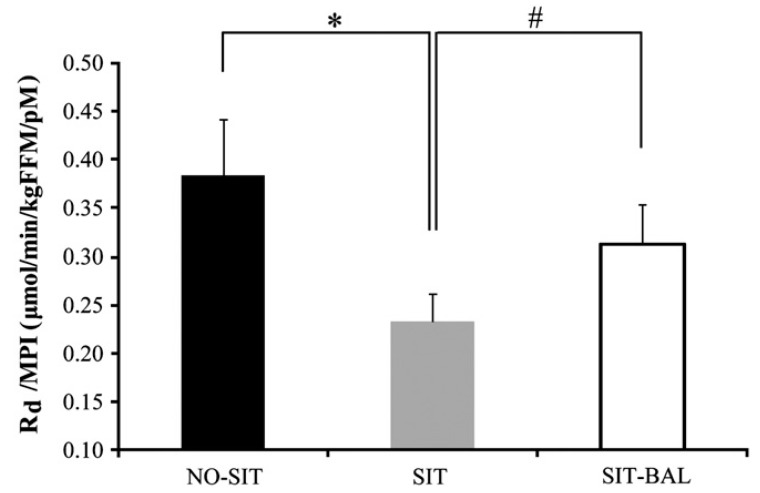
Whole-body insulin action (R_d_/MPI) assessed during the continuous infusion of glucose across the three conditions. Values are mean ± SEM. MPI, mean of 140-, 145-, and 150-min plasma insulin concentrations. Insulin action was 39% lower in SIT compared with NO-SIT (* *p* < 0.001). Insulin action was 26% lower in SIT compared with SIT-BAL (^#^
*p* = 0.04). Insulin action was 18% lower in SIT-BAL compared with NO-SIT (*p* = 0.07). Republished with permission of Canadian Science Publishing from Stephens et al., 2011 [49]; permission conveyed through Copyright Clearance Center, Inc.

**Table 1 nutrients-11-02230-t001:** Values are means ± SE.

	Energy Deficit				Energy Balance			
	Pre	Post	Δ	95% CI	Pre	Post	Δ	95% CI
**Triacylglycerols (mg/dL)**	117 ± 23	98 ± 15	−19	−49, 11	133 ± 39	126 ± 27	−7	−36, 22
**Total cholesterol (mg/dL)**	203 ± 8	192 ± 9	−11	−24, 1	178 ± 10	173 ± 10	−5	−21, 11
**HDL (mg/dL)**	52 ± 3	53 ± 3	+1	−5, 3	51 ± 4	46 ± 5 *	−5	−9, −1
**LDL (mg/dL)**	127 ± 6	120 ± 6	−7	−19, 4	100 ± 14	102 ± 14	+2	−15, 19
**Adiponectin (ng/dL)**	9.95 ± 1.4	10.46 ± 1.2	+0.5	−0.21, 1.23	6.47 ± 1.4	6.51 ± 1.3	+0.04	−1.21, 1.29
**CRP (mg/L)**	3.8 ± 0.6	3.1 ± 0.6	−0.7	−2.25, 0.83	4.9 ± 1.7	5.0 ± 1.6	+0.1	−0.99, 1.24
**Leptin (ng/mL)**	16.2 ± 2.1	13.6 ± 2.2 *	−2.6	−5.16, 0.01	14.0 ± 3.1	14.0 ± 3.2	−0.1	−1.09, 0.96

CRP: C-reactive protein. * Significantly different from pretraining, *p* < 0.05. Adapted from Black et al., 2005 [22]. HDL = High-density lipoprotein, LDL = Low-density lipoprotein, CRP = C-reactive protein.

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
