# Peer review of "Accounting for the Nutritional Context to Correctly Interpret Results from Studies of Exercise and Sedentary Behavior"

_nutrients, 2019, doi:10.3390/nu11092230_

Round 1

Reviewer 1 Report

Major Comments

-I would suggest making different section within the text and giving some sort of structure to the manuscript so that it will be easier for readers to follow. For example, it will be really useful to talk about the relationship between dietary components such proteins, fats, carbohydrates, energy and etc. with exercise in different sections as some readers may be interested in some part of this review paper depending upon their area of interest and they can find those parts easily. Also, that would be useful to have different section for exercise and sedentary behavior.

-The authors should support their statements with some references from previous studies. For example, in lines 68-92 only 1 reference has been used without providing support to the rest of discussions.

-Bringing some studies from animal models in the review paper and talking about the mechanisms by which dietary components affect the outcome of exercise on energy balance would add value to the paper as other than clinicians, basic obesity scientists can also benefit from those discussions.

Minor comments

Line 23: Delete extra “from the”

Line 26: Delete the “,” and full stop.

Line 49: Delete the extra “fat”

Line 68-69: Please include some of these references here.

Line 100: Remove the underline

Line 144: Remove the underline

Line 171-172: The reference is missing

Author Response

Major comments:

As suggested, we created text sections/headers for readability

We added references to line 68-92

We have added several studies in animals to give the reader a little insight into the mechanisms by which carbohydrate intake blunts the insulin sensitivity response to exercise.

Minor comments:

Line 23 – deleted extra “from the”

Line 26 -deleted the “  ” and full stop

Line 49 – replaced “fat, carbohydrate, and fat” with “protein, carbohydrate, and fat”

Line 68-69 – added references

Line 100 – removed underline

Line 144 – removed underline

Line 171-172 – the reference is Stephens et al. [47]. Not referenced in the first sentence but referenced in the second sentence

Reviewer 2 Report

Braun and Newman have reviewed the evidence for the importance of nutritional context, especially energy balance, for the outcomes of exercise studies. This review is important, timely, and well-written. The authors should be congratulated on their efforts.

Major points:

This review is important because it clearly demonstrates the potential for changes in diet, especially energy balance, to confound findings of studies. This is a feasible explanation for the wide differences found within and between studies of the effects of exercise on, for example, insulin sensitivity. It is of vital importance that nutrition scientists better control their experiments, because their results are often reported in the media, and are used as a basis for lifestyle changes by real people.

The authors chose a single outcome – insulin sensitivity – as the focus for their work. This is not a flaw, but a strength, as they are demonstrating a concept and have used insulin sensitivity as an example. This keeps the review to a reasonable length and allows us to ponder how the effect of energy balance may have confounded studies measuring all outcomes.

The quality of the reproduced figures is often quite poor. If the original artwork cannot be obtained at sufficient resolution, it might be worth remaking the figures.

Minor points:

The writing was engaging and very well done. The authors occasionally use some odd words, which I will outline below. It is very rare that there are so few errors in a manuscript that I can actually list them, but they can be found here:

Line 26: “disability, A buffet..” should be “disability. A plethora…” (or something – buffet is an odd choice. Line 41: “The converse is also true, exercise…” should be “The converse is also true; exercise” (make friends with the semicolon; it will serve you well) Line 64: “…profound effects (and best evidence) comes from…” should be “…come from” (the word effects is plural). Line 65: the word “resistance” here is unclear – the sentence should be rewritten. Line 73: delete “of attention” Line 94: “minor energy deficit or surplus” should be plural; i.e. “deficits or surpluses” Line 100: “intake” use italics instead of underlining it Line 115: “is” should be “in” Line 144: “overcompensated” – use italics for the “over” part Line 173: at least one word is missing from this sentence. Did you mean “Stephens et al. tested WHETHER the impact of a day of inactivity would reduce insulin…”? Otherwise this sentence does not make sense. Line 198: the full stop after “energy balance” should be a comma Lines 222 to 223: “Given the reality that high costs and administrative burden” should be “Given the reality that high costs and THE administrative burden” Line 237: “fore” – this word is odd – I’m not sure what you mean by this. Generally: just check that there is only one space between a full stop and the new sentence.

Author Response

Thank you for constructive comments and good suggestions. As noted below, we have responded to each comment and made the appropriate changes to the text.

Major comments:

Edited figure 1 and figure 3 – increased sharpness of figures

Minor comments:

Line 26 – changed “buffet” to “many”

Line 41 – changed comma to semi-colon

Line 64 – changed comes to come

Line 65 -removed “resistance”

Line 73 – removed “of attention”

Line 94 – changed deficit and surplus to deficits and surpluses

Line 100 – removed underline, changed to italics

Line 115 – changed “is” to “in”

Line 144 – removed underline, changed to italics

Line 173 – added “whether”

Line 198 – removed full stop, added comma

Line 222-223 – added “the” in front of administrative burden

Line 237 – changed “fore” to “fire” (i.e. add fuel to the "understanding context" fire)

General – removed extra spaces between full stops and the subsequent sentence

Round 2

Reviewer 1 Report

The authors have added headings to improve the readability, which is an overall improvement. However, the authors have not highlighted the text for the parts that they have possibly made changes. In response to the reviewer's request for adding new citations and elaborating the discussion on nutrients and the mechanisms by which dietary components affect the outcome of exercise on energy balance, the authors have stated that it was done without providing any explanations or highlighting the text for the reviewers to evaluate the revised version of the article. The authors have mentioned "We have added several studies in animals to give the reader a little insight into the mechanisms by which carbohydrate intake blunts the insulin sensitivity response to exercise". How about other major dietary components such as proteins and fats? Is there a reason that only dietary carbohydrates are discussed while it is not clear where those changes have been applied in the paper and what new citations have been added. Therefore, it is almost impossible for the reviewer to evaluate the current versions of the paper as I'm not convinced if my previously requested revisions have been appropriately applied.

Author Response

Our apologies for the lack of clarity regarding the revisions. We have highlighted the modified/added text in the current version. Additionally, we have included studies that discuss the manipulation of dietary fat and the effect on insulin sensitivity. To the best of our knowledge, there is no research on the consequences of protein surplus or deficit following exercise on insulin sensitivity. 

Round 3

Reviewer 1 Report

The manuscript has been improved and new changes were seen by the reviewer now. New citations should be also highlighted and the references cross-checked again for the whole manuscript.

Author Response

Thank you! We highlighted the added references within the text and in the references section. We have also cross-checked the references again for accuracy.